# Reactive Oxygen Species-Mediated Cytotoxicity in Liver Carcinoma Cells Induced by Silver Nanoparticles Biosynthesized Using *Schinus* *molle* Extract

**DOI:** 10.3390/nano12010161

**Published:** 2022-01-03

**Authors:** Waleed Ali Hailan, Khalid Mashay Al-Anazi, Mohammad Abul Farah, Mohammad Ajmal Ali, Ahmed Ali Al-Kawmani, Faisal Mohammed Abou-Tarboush

**Affiliations:** 1Department of Zoology, College of Science, King Saud University, P.O. Box 2455, Riyadh 11451, Saudi Arabia; hilan66@gmail.com (W.A.H.); kalanzi@ksu.edu.sa (K.M.A.-A.); ahmadkomani@gmail.com (A.A.A.-K.); ftarbush@ksu.edu.sa (F.M.A.-T.); 2Department of Botany & Microbiology, College of Science, King Saud University, P.O. Box 2455, Riyadh 11451, Saudi Arabia; alimohammad@ksu.edu.sa

**Keywords:** HepG2, *Schinus molle*, silver nanoparticles, biosynthesis, autophagy, apoptosis

## Abstract

Hepatocellular carcinoma (HCC) is the most common primary liver malignancy and is ranked as the third most common cause of cancer-related mortality worldwide. *Schinus molle* (*S. mole*) L. is an important medicinal plant that contains many bioactive compounds with pharmacological properties. The role of *S. molle* leaf extract in the biosynthesis of silver nanoparticles (AgNPs) was determined. The biosynthesized AgNPs were thoroughly characterized by UV–vis spectrophotometry, transmission electron microscopy (TEM), X-ray diffraction (XRD), and dynamic light scattering (DLS) techniques. Furthermore, the cytotoxic effect of the biosynthesized AgNPs using *S. molle* (SMAgNPs) against HepG2 liver cancer cells was investigated. Reactive oxygen species generation, apoptosis induction, DNA damage, and autophagy activity were analyzed. The results clearly showed that the biosynthesized silver nanoparticles inhibited the proliferation of HepG2 by significantly (*p* < 0.05) inducing oxidative stress, cytotoxicity, DNA damage, apoptosis, and autophagy in a dose- and time-dependent manner. These findings may encourage integrating the potential of natural products and the efficiency of silver nanoparticles for the fabrication of safe, environmentally friendly, and effective anticancer agents.

## 1. Introduction

Hepatocellular carcinoma (HCC) is the most common primary liver malignancy, representing approximately 75–85% of cases [1]. Additionally, HCC is ranked as the second most common cause of cancer-related mortality among men and the sixth most common cause of cancer-related death among women [2]. The incidence and mortality of liver cancer have continued to rise despite advances in new technologies, prevention techniques, and screening for both diagnosis and treatment [3]. Cancer treatment strategies include radiotherapy, chemotherapy, and surgery and can be implemented to improve a patient’s quality of life. However, there are many major problems that must be handled during the cancer treatment process, one of them being the side-effects that are related to conventional medical treatment strategies [4,5].

Recently, the nano-biotechnology applications have revealed novel strategies for the treatment and diagnosis of cancer [6]. Nowadays, the application of silver nanoparticles (AgNPs) as an auspicious anticancer entity is becoming versatile and widespread [7]. AgNPs have demonstrated low toxicity in humans as well as diverse application in in vitro and in vivo studies [8,9]. AgNPs have been synthesized using multiple physico-chemical techniques. Recently, various plant materials, including leaves, fruits, stems, roots, seeds, and their extracts, have been exploited as an eco-friendly, cost reductive, and nontoxic approach for the effective biosynthesis of AgNPs [10,11,12]. 

*Schinus molle* L. is one of the most important medicinal plants, and it has shown to be effective when used as an antifungal antibacterial, antiviral, topical antiseptic, analgesic, anti-inflammatory antispasmodic, and antioxidant as well as antitumoral agent [13,14,15]. Its diverse range of applications can be attributed to it containing many bioactive compounds with various pharmacological properties, such as beudesmol, myrcene, a-phellandrene, elemol, limonene, and caryophyllene [16,17]. Since the high levels of phenolic components that are found in *S. molle,* especially in the aqueous extract fraction, play a vital role in the biosynthesis of silver nanoparticles [18], these compounds could act as reducing and capping agents. It is worth noting that the plant source is what identifies the properties of the biosynthesized AgNPs [19]. Considering these advantages, the present study aimed to use the aqueous extract from *S. molle* leaves in order to obtain silver nanoparticles. Then, the biosynthesized AgNPs were thoroughly characterized by UV–vis spectrophotometry, TEM (transmission electron microscopy), dynamic light scattering (DLS), and XRD (X-ray diffraction) techniques. The anticancer potential of the biosynthesized AgNPs that were obtained from *S. molle* (SMAgNPs) against HepG2 liver cancer cells was investigated by evaluating intracellular ROS generation, apoptosis induction, DNA damage, and autophagy activity. 

## 2. Materials and Methods

### 2.1. Biosynthesis and Characterization of Silver Nanoparticles

Fresh leaves from *Schinus molle* were collected from Riyadh, Saudi Arabia, and these leaves were identified and processed using the methodology reported by Farah et al. [10]. Briefly, the collected leaf materials were rinsed, air- dried, and then powdered and boiled in Milli-Q deionized water (Millipore, Billerica, MA, USA) for 20 min. The aqueous extract was separated by means of filtration with Whatman No. 1 filter paper (Maidstone, UK). An aqueous solution (1 mM) of silver nitrate (AgNO_3_) was prepared in a 250 mL Erlenmeyer flask, and then an aqueous extract was added in order for the particles to reduce into silver ions. The reaction (1 mL aqueous leaf extract + 99 mL 1 mM AgNO_3_) was conducted in darkness in order to avoid the photoactivation of AgNO_3_ at room temperature. The change of the mixture solution from slightly yellowish colour to brown indicated that AgNPs had synthesized. The mixture solution was centrifuged at 10,000 rpm for 10 min, and the supernatant was removed. The pellet (AgNPs) was collected and washed twice with Milli-Q deionized water, and it was then air- dried in an incubator at 50 °C. The powder of the biosynthesized AgNPs was kept in tightly sealed tubes in dark and ambient conditions until it was needed for further characterization and applications.

The colour change of reaction mixture was observed via direct visualization, and its UV spectra were recorded using an ultraviolet-visible spectrophotometer (BioTek, Winooski, VA, USA) at wavelengths that ranged between 300 and 700 nm. Further, to confirm the size and shape of the biosynthesized AgNPs, a trace amount of a AgNPs sample was added to a carbon-coated copper grid, images were then captured at an accelerating voltage of 120 kV by means of a transmission electron microscope (TEM) (JEOL, Tokyo, Japan). Photographs from several fields were captured, and more than 100 numbers of SMAgNPS were analysed for average size distribution. X-ray diffraction patterns of the biosynthesized nanoparticles were measured using an XRD device (PANalytical, Almelo, the Netherlands) with a voltage of 40 kV and a Cu-Kα (λ = 1.5418 Å) radiation source 2θ that ranged from 32–80°. The size distribution profile and the zeta potential were measured through dynamic light scattering (DLS) analysis, which was conducted on a Malvern Zetasizer Nano ZS instrument (Malvern Instrument Limited, Malvern, UK).

### 2.2. Cell Culture 

Human liver cancer cell lines HepG-2 (ATCC-HB-8065) were procured from American Type Culture Collection (ATCC^®^, Manassas, VA, USA). Dulbecco’s modified Eagle’s medium (DMEM) supplemented with 10% fetal bovine serum (FBS) and 1% antibiotics (Invitrogen™ GIBCO^®^) was used to culture the cells. The cells were incubated at 37 °C with 5% CO_2_ as an adherent monolayer in a CO_2_ incubator (BINDER^®^, Tuttlingen, Germany). When the cells reached 80–90% confluence, they were transferred to a new flask by means of the trypsinization process.

### 2.3. MTT Cytotoxicity Assay

An MTT test, which reflects the number of viable cells through the evaluation of the NAD(P)H-dependent cellular oxidoreductase enzyme activity, was performed to evaluate the cytotoxicity [20]. In brief, the HepG2 cell were harvested, counted, and seeded in 96-well plates at a density of 10^4^ cells/well, and the cells were then incubated overnight. The exponential growth of the cells at 70–80% confluence was treated with different concentrations of SMAgNPs and *S. molle* extract (10–500 µg/mL), and they were then further incubated for 24 h at 37 °C. Negative control wells (wells without treatment) were also maintained for comparison. The assay was performed following the manufacturer’s recommendations as per the assay kit (Promega, Madison, WI, USA). The optical density (OD) of each well was measured using a synergy microplate reader (BioTek^®^, Winooski, VA, USA) at 550 nm. The cell viability percentage was then calculated by assuming that the cell viability for the negative control was 100%. The half maximal inhibitory concentration (IC_50_), which was derived from a dose–response curve, was determined. Then, three different doses of SMAgNPs that were below the IC_50_ value were selected and were used in all of the subsequent experiments. The IC_50_ value in cell viability test for any drug-like compound is the concentration at which 50% growth of the cell is inhibited.

### 2.4. Lactate Dehydrogenase Cytotoxicity Assay

A lactate dehydrogenase (LDH) test was employed to evaluate the cytotoxicity based on the quantification of plasma membrane impairment in order to measure the number of lysed cells in the presence of toxic materials [21]. A commercial LDH assay kit (Abcam, Cambridge, UK) was use by following the manufacturer’s instructions. Briefly, 1 × 10^4^ cells per well were cultured in 96-well plates overnight. The cells were treated with three different concentrations of SMAgNPs and *S. molle* extract (20, 40, and 60 μg/mL) at 37 °C for 12 and 24 h. After the completion of desired duration, the culture plate was centrifuged at 1000 rpm for 10 min. Then, 50 μL of cell culture supernatant was transferred from each well to a fresh 96-well plate, and 100 μL of the freshly prepared LDH reaction mixture was added to each well. After incubation for 30 min at room temperature in the dark, the absorbance was measured at a wavelength of 450 nm using a synergy microplate reader (BioTek, Winooski, VA, USA). The LDH content was expressed as a percentage based on the number of control cells, which was assumed to be 100%.

### 2.5. Cytomorphological Changes Analysis

The HepG2 cells were cultured on a 6-well plate at a density of 1 × 10^5^ cells/well. Morphological changes were examined to investigate the alterations that were induced by the SMAgNPs in the cells that had been treated with 20, 40, and 60 μg/mL for 24 h, and these changes were observed under an inverted microscope equipped with a digital camera (Olympus IX51, Tokyo, Japan) at 100× magnification.

### 2.6. Measurement of Intracellular Reactive Oxygen Species (ROS) Using Carboxy-H_2_DCFDA

For the estimation of the ROS in the HepG2 cells, a commercial kit (Image-iT™ LIVE Green ROS Detection Kit, Molecular Probes, Eugene, OR, USA) was used. Briefly, 1 × 10^4^ cells/well were seeded in 96-well culture plates (black-bottomed) that were then exposed to three different concentrations of SMAgNPs (20, 40, and 60 μg/mL) at 37 °C for 12 and 24 h. Then, the cells were washed with PBS (phosphate-buffered saline) and were incubated with a working solution of carboxy-H_2_DCFDA (25 μM) in phosphate-buffered saline (PBS) for 30 min. The fluorescence intensity was detected at an excitation wavelength of 485 nm and at an emission of 528 nm using a synergy microplate reader (BioTek, Winooski, VA, USA). The values were expressed as a percentage of the fluorescence intensity relative to the control wells. Correspondingly, the qualitative method using fluorescence microscopic imaging was executed after seeding the cells on a coverslip-loaded six-well plate and was processed as above. Then, images were captured using a compound microscope (Olympus BX41, Tokyo, Japan) that was equipped with an appropriate fluorescence filter and a charge-coupled device (CCD) camera.

### 2.7. Apoptotic Morphological Changes by Acridine Orange–Ethidium Bromide (AO-EB) Dual Staining Method

To detect the apoptosis-associated cell membrane changes and nuclear morphological alterations in HepG2 cells, the cells were seeded on a coverslip-loaded six-well plate with 1 × 10^5^ cells/well. Then, the cells were treated with the prospective SMAgNP treatments (20, 40 and 60 μg/mL) at 37 °C for 12 and 24 h. The cells were incubated with a working solution that contained 100 µg/mL AO and 100 µg/mL EB (AO-EB, Sigma, St. Louis, MO, USA) in PBS for 10 min followed by observation under a compound fluorescence microscope (Olympus BX41, Tokyo, Japan) that had been equipped with a suitable fluorescence filter (488 nm excitation, 525 nm emission) and a CCD camera. A total of 300 cells were examined in each sample based on the fluorescence emission and the morphological features of the chromatin condensation/fragmentation that was present on the cells [12,22].

### 2.8. Flow Cytometry Analysis of Apoptosis

A commercial kit (FITC Annexin V Apoptosis Detection Kit) supplied by BD Pharmingen™, (CA, USA) was used according to the instructions provided in the kit to evaluate the induction of apoptosis in the HepG2 cells. Briefly, the HepG2 cells were grown and exposed to the selected doses of SMAgNPs (20, 40, and 60 μg /mL) for 12 and 24 h. Cells were resuspended in 1× binding buffer and were incubated with annexin V–FITC and propidium iodide (5 µL each) for 20 min in the dark. All of the samples were immediately analyzed in a flow cytometer (BD FACS Calibur^®^), and the data were evaluated using Cell Quest^®^ Pro software (BD).

### 2.9. Detection of Autophagy by Acridine Orange Staining

The AO staining method is a quick, accessible, and reliable way to assess the formation of acidic vesicular organelles (AVOs) as a hallmark of autophagy induction using fluorescence microscopy. In short, the HepG2 cells were seeded on a cover slip-loaded six-well plate and were then treated with the prospective doses (20, 40, and 60 μg/mL) for the desired durations (12 and 24 h). After that, the cells were incubated with 5 μg/mL of the working solution used for AO for 15 min in dark conditions. The stained cells were washed with PBS and were then mounted on glass slides and were observed under a compound microscope (Olympus BX41, Tokyo, Japan) that had been fitted with a fluorescence attachment and a CCD camera.

### 2.10. Alkaline Single Cell Gel Electrophoresis

DNA damage was detected using the single-cell gel electrophoresis technique (comet assay) proposed by Singh et al. [23], with some modifications [10]. The HepG2 cells were grown in a six-well culture plate and were then treated with the different doses (20, 40 and 60 μg/mL) for 12 and 24 h. After the cells had undergone gel electrophoresis, the slides were immersed in neutralization buffer (0.4 M Tris-HCl, pH 7.5) and were then stained with ethidium bromide (20 μg/mL) for 5 min. The slides were analyzed under a fluorescence microscope (Olympus BX41, Tokyo, Japan) using the appropriate filter settings (excitation wavelength of 515–560 nm and emission wavelength of 590 nm). For a quantitative evaluation of the DNA damage, 200 cells from each category were randomly selected and were subjected to image analysis using Comet Assay IV software (Perceptive Instruments, Suffolk, UK). 

### 2.11. Statistical Analysis

All of the experiments were performed three times with sufficient replicates, and the data were presented as mean ± standard deviation (SD). Statistical analyses were performed with SPSS software (Version 22, SPSS Inc., Chicago, IL, USA). One-way ANOVA was used to determine the significance of the data. *p*-values < 0.05 were considered statistically significant.

## 3. Results and Discussion

### 3.1. Biosynthesis and Characterization of AgNPs

With an ever-increasing number of nanomaterials that have potential in oncology applications being available to researchers, the choice of which material to use for a specific type of cancer now is an important question [24]. Silver nanoparticles have an advantage over other biomedical nanomaterials, as they exhibit particularly unique physical, chemical, optical, and biological properties. Therefore, silver nanoparticles can serve as therapeutic platforms in many biomedical applications that are included but not limited to anticancer therapeutic agents for hepatocellular carcinoma treatment [8]. The choice of the leaf extract from *S. molle* was based on the presence of a variety of phytochemical compounds such as phenols, amino acids, flavones, etc., [25,26]. These molecules were expected to self-assemble and cap the metal nanoparticles that formed in their presence, thereby inducing some form of shape control during metal ion reduction [27].

At the onset of AgNPs synthesis, the colour of the reaction mixture began to change within 20 min of addition of the *S. molle* leaf extract, changing from colorless to brown (Figure 1A), indicating the biosynthesis of the AgNPs. The reduction of the silver ions to silver nanoparticles was emphasized through different techniques. The primary and most common technique that is used to characterize SMAgNPs is UV-Vis Spectroscopy analysis, which revealed that the surface plasmon resonance (SPR) of Ag occurred at around 412 nm (Figure 1B), which is a typical characteristic feature of AgNPs. 

TEM analysis showed that the SMAgNPs had a spherical shape, with size ranged from 16.5 to 38.5 nm (Figure 2A). The nanoparticles appeared to be evenly separated and well dispersed. Figure 2B shows the frequency of the average size distribution of the SMAgNPs. The bioactive constituents in the *S. molle* leaf extract may act as reducing and capping agents during the biosynthesis of the AgNPs, providing them long term stability [25]. As shown in Figure 3, the XRD patterns of the SMAgNPs powder exhibited diffraction peaks with 2θ values of approximately 8.05, 44.11, 64.43, and 77.36, corresponding to 111, 200, 220, and 311, respectively, clearly indicating the formation of face-centered cubic (FCC) crystalline silver nanoparticles. Notably, the additional peaks belong to the organic compounds in the extract that were potentially responsible for the reduction and stabilization of the silver ions. The size distribution profile of the biosynthesized NPs was recorded as 135.8 nm using the DLS method (Figure 4A). The polydispersive index (PdI) was observed as 0.790. The zeta potential was observed to be −24.9 mV (Figure 4B), showing that the nanosuspension had high stability. The negative zeta potential was due to negative charge accumulation at the surface, cores of the particles demonstrated positive charge accumulation.

Therefore, in the present study, the AgNPs were synthesized successfully using *S. molle* leaf extract as an active reducing agent. The colour change to dark brown occurred after the silver particles had increased in size. The absorbance of that colour change, which was measured by UV-spectroscopy, revealed surface plasmon resonance at 400−420 nm, confirming the reduction of Ag+ ions to metallic Ag [28]. The reduction of the silver ions could be attributed to the abundant presence of reductive phenolic compounds in the *S. molle* leaf extract that enhance the AgNPs synthesis in ambient conditions [27]. These results were verified by transmission electron microscopy (TEM), where the micrographs showed that the SMAgNPs had a spherical shape and that they were in the size range of 16.5–38.5 nm. This finding was supported by a previous report by Ratan et al. [29], which indicated that most of the nanoparticles that are synthesized from plant sources are spherical in shape The crystalline structure of the SMAgNPs in the present study was confirmed by analyzing the XRD pattern, which was shown to match the standard reference for FCC structure determined by the Joint Committee of Powder Diffraction Standards (JCPDS) card No-04-0783.

### 3.2. Cytotoxicity Assessment of SMAgNPs 

The cell viability assay is one of the most important methods for toxicology analysis and explain the cellular response to toxic materials; it can provide information on cell proliferation, death, and metabolic activities [30]. In the present study, the MTT assay was used to investigate the potential antiproliferative effect of SMAgNPs at the different concentrations of 20, 40, and 60 µg/mL. The MTT results that are depicted in Figure 5 show a significant (*p* < 0.05) decrease in the viability of the HepG2 cells that had been treated with SMAgNPs in a concentration-dependent manner in the range of 12–94% with 500–10 μg/mL concentrations when compared to the control wells (untreated), while those that were treated with *S. molle* extract treated alone demonstrated decreased cell viability in the range of 34–96% across the above concentrations. The IC_50_ values of the SMAgNPs and the *S. molle* extract were estimated to be 92 μg/mL and 135 μg/mL, respectively, after 24 h of treatment. The SMAgNPs exerted higher toxicity in the HepG2 cells. Accordingly, three different concentrations that were less than the IC_50_ value (20, 40, 60 μg/mL) were applied in subsequent experiments to study the anticancer activity of SMAgNPs in HepG2 cells. SMAgNPs exerted higher toxicity in HepG2 cells when compared to the effect of *S. molle* extract. It appears that there were additive effects of phytochemicals present in extract when attached to the AgNPs. In general, the IC_50_ value of different silver nanoparticles formulations differ widely according to the size of NPs, cell lines type, stabilizing agent present in the extract and its concentrations. Previously, the IC_50_ value of AgNPs biosynthesized from *adenium obesum* (217 μg/mL) and *Ochradanus arabicus* (100 μg/mL) was reported in breast cancer cells [10,12].

### 3.3. LDH Release Assay

SMAgNP-induced cytotoxicity was confirmed quantitatively by measuring the LDH activity that was released into the extracellular medium due to membrane damage. Figure 6 shows the results of the LDH cytotoxicity assay, which revealed that the SMAgNPs induced a dose-dependent release of LDH in HepG2 cells. A significant (*p* < 0.05) increase of 78% and 140% LDH was registered in the highest concentration group (60 µg/mL) after 12 h and 24 h, respectively, indicating a disruption of the cell membrane structure. A similar effect was also observed after *S. molle* extract treatment. LDH release also increased from 136% to 145% in 60 µg/mL treatment group after 12 h and 24 h.

Therefore, the LDH data were in line with the MTT cell viability results. Our findings revealed that the cell proliferation percentages were significantly reduced in a dose- and time-dependent manner. The results in this study are corroborated by the results of other investigators reporting that AgNPs significantly affected cell viability and cell membrane integrity [31,32].

### 3.4. The Cytomorphological Alterations 

The morphological changes assessment resulted in evident changes in HepG2 cells after exposure to SMAgNPs for 24 h. The inverted microscope images that are shown in Figure 7 reveal clear changes in the morphology of the HepG2 cells in all three of the concentrations that were used, especially at the highest ones, compared to the control. These alterations mainly included changes to the typical shape, cell shrinkage, cell detachment, and a decline in cell density. At the same time, there were not any remarkable morphological changes that were observed in the control cells. These results are in accordance with our previous findings [10,33]. 

### 3.5. Estimation of Intracellular ROS Generation

Although ROS are generated in living systems as a normal part of metabolic activities, it is well known that the over production of ROS is associated with the activation of cell apoptosis, the overproduction of ROS has been tied especially to anticancer drugs [34]. The treatment of cells with SMAgNPs resulted in the induction of ROS production. As shown in Figure 8A–H, there was an increase in the ROS level in a concentration-dependent manner. A clear enhancement of the green fluorescence intensity is clearly observed compared to the controls (Figure 8A,E). The maximum level of ROS generation was induced by the highest concentration (60 µg/mL) that was used, whereas the minimum level of ROS production was closely associated with the lowest concentration of 20 µg/mL compared to the control (Figure 8A–H). These results were corroborated by the fluorescence spectroscopy measurements, which indicated that the HepG2 cells were the most affected with the 60 µg/mL SMAgNPs concentration, which exhibited a statistically significant (*p* < 0.05) increase in ROS accumulation (145% and 176 %) when compared to the respective control after 12 h and 24 h, respectively (Figure 8I).

Many previous studies have provided strong evidence for a link between AgNP-mediated production of ROS, the subsequent generation of oxidative stress, and cytotoxicity [35,36]. On the other hand, there is conflicting evidence regarding the dependency of AgNP induced cytotoxicity on ROS [37]. The exact mechanism through which SMAgNPs generate ROS is not clear. Moreover, the mechanism of ROS generation is not specific to the liver cells. However, some of the studies have shown that cancer cells are more sensitive in generating ROS than in normal cells [38]. This is due to the tumor microenvironment which have low pH (acidic) and have high requirement of energy for oxidative metabolism.

### 3.6. Apoptotic Morphological Changes by (AO-EB) Dual Staining Method 

Apoptosis detection in the treated HepG2 cells was observed via the morphological changes in the cell shape and chromatin condensation as well as by nuclear fragmentation coupled with DNA integrity, which are hallmarks of apoptosis [39]. The images that were captured using the fluorescence microscope for the detection of apoptotic morphological changes by AO-EB staining showed that the untreated control cells appeared to be mostly green and that they tended to have a regular shape and intact nuclear architecture (Figure 9A,E), whereas the images of the treated cells demonstrated that many cells had undergone apoptosis, which was identified by orange fluorescence and morphological alterations such as apoptotic bodies, chromatin condensation, and fragmentation. Otherwise, some cells lost their membrane integrity, especially at the 60 μg/mL concentration, and these cells were stained with red, representing necrosis (Figure 9D,H). The quantification of the apoptotic, necrotic, and viable cells displayed that the percentage of viable cells decreased significantly (*p* < 0.05) with SMAgNPs treatment. The highest percentage of apoptotic cells (15.4% and 24.7%) was recorded at the highest concentration of 60 µg/mL after exposure for 12 h and 24 h, respectively (Figure 9I).

### 3.7. Annexin-V/PI Apoptosis Assay

The above findings were emphasized by the Annexin-V/PI apoptosis assay results that were obtained via flow cytometry analysis. The results of the apoptosis quantification in HepG2 cells using flow cytometry were presented in dot plots (Figure 10A), which showed a significant increase in apoptosis levels in a dose- and time-dependent manner. The highest rates of early apoptosis were 9.3% and 11% at the 60 µg/mL concentration after 12 and 24 h, respectively, whereas the levels of late apoptosis were 11.4% and 10.4% at the highest concentration (60 µg/mL) for the 12 h and 24 h durations, respectively, compared to the controls (Figure 10B). On the other hand, treatment with the SMAgNPs induced slightly elevated necrosis in a time- and concentration-independent manner.

Apoptosis plays an essential role in a wide variety of different biological systems, including in the normal cell cycle, anticancer mechanisms, the immune system, embryonic development, morphological changes, and chemical-induced cell death [40]. Many plant-derived compounds show promise for use in antitumor therapy via the activation of the apoptosis pathways [41]. In this context, the over-production of the intracellular ROS that was induced by the SMAgNPs in the HepG2 cells may play an important role in the anticancer activity of biosynthesized AgNPs by elevating oxidative stress, cytotoxicity, and apoptosis [42]. These findings are in accordance with results that were previously reported by Kim et al. [43] and Ahmadian et al. [44].

### 3.8. Detection of Autophagy by Acidic Vesicular Organelles (AVOs)

Although autophagy is linked to pathologic conditions such as cancer, it is being studied as a therapeutic target [45]. A number of studies have shown that autophagy is a cellular death mechanism as well as a response to various anticancer therapies in many kinds of cancer cells [46,47]. There is an increasing body of evidence showing a link between ROS and autophagy, in which ROS generation could trigger autophagy [48]. The fluorescence microscopic images presented in Figure 11A–H clearly show elevated acidic vesicular organelles (AVOs, an indicator for autophagy) in the treated HepG2 cells in a time- and concentration-dependent manner. The control (untreated) cells exhibited limited AVOs, which could be inferred through the amount of green fluorescence-stained cytoplasm and the negligible instances of orange fluorescence. The HepG2 cells that were treated at higher concentrations (60 µg/mL, Figure 11D,H) contained more AVOs than those that were treated at lower concentrations (20 µg/mL, Figure 11B,F) after the same length of exposure to the SMAgNPs. Notably, the treatment showed plenty of AVOs after 24 h compared to after 12 h at the same concentration.

In the present study, the occurrence of autophagy coincided with the oxidative stress, and this may indicate a correlation between oxidative stress resulting from ROS accumulation and autophagy [49]. Autophagy is a double-edged sword that induces cancer growth progression, especially in the early stages of the disease, but that also works to suppress tumors depending on the context and cell type [50]. 

### 3.9. DNA Damage

The DNA damage was measured as the DNA tail length in the HepG2-treated cells compared to in the control cells. The fluorescence microscope images of the comet tails in the SMAgNP-treated and untreated HepG2 cells are presented in Figure 12A–D, which shows images that were taken after 24 h of exposure. It was evident that that the number of cells with comet-like tails increased remarkably as the SMAgNPs concentrations increased. The quantitative estimation of the comet tail length is presented in Figure 12E and demonstrates a significant (*p* < 0.05) time- and concentration-dependent increase in the DNA damage present in the treated cells. The highest DNA damage with the maximum tail length (32.6 and 40.4 μm) was recorded at the 60 µg/mL SMAgNPs concentration after 12 h and 24 h of exposure compared to the undamaged control cells. Due to the elevated levels of oxidative stress and cytotoxicity, exposure to the SMAgNPs induced single-strand breaks in the DNA and led to an increase in comet tail development, which is an indicator of DNA damage [51].

The cytotoxicity of the AgNPs against human liver HepG2 cells could be attributed to the oxidative stress and cell apoptosis taking place in the treated HepG2 cells. Moreover, cytotoxicity was observed in a concentration-dependent manner, and it could have been induced by various mechanisms that are related to mitochondrial injury [52,53]. The mechanism underlying the cytotoxic and antitumor effects of silver nanoparticles is still unknown. The findings that were presented here showed that the treatment of HepG2 cells with SMAgNPs induced cytotoxicity in a ROS-dependent manner that was also accompanied by apoptosis and autophagy. These findings provide a novel approach for the application of biosynthesized AgNPs towards cancer therapy [47,54]. It has been suggested that the mechanisms that are involved in ROS generation, oxidative stress, DNA damage, apoptosis, and autophagy are the most commonly reported mechanisms of AgNP toxicity [10,11,12,42,52].

## 4. Conclusions

In the present study, the medicinal plant *S. molle* was used for the biosynthesis of silver nanoparticles. That was conducted due to the presence of active compounds in the aqueous extract from the leaves of this plant that are able to act as capping and reducing agents for silver ions. The results clearly emphasized the strong ability of silver nanoparticles that had been biosynthesized from *S. molle* to inhibit the proliferation of HepG2 cells, the process of which might be mediated through the induction of oxidative stress, cytotoxicity, apoptosis, and autophagy. This promising approach may encourage researchers to integrate the potential of natural products and the efficiency of silver nanoparticles for the creation of safe, environmentally friendly, and effective therapies for liver cancers.

## Figures and Tables

**Figure 1 nanomaterials-12-00161-f001:**
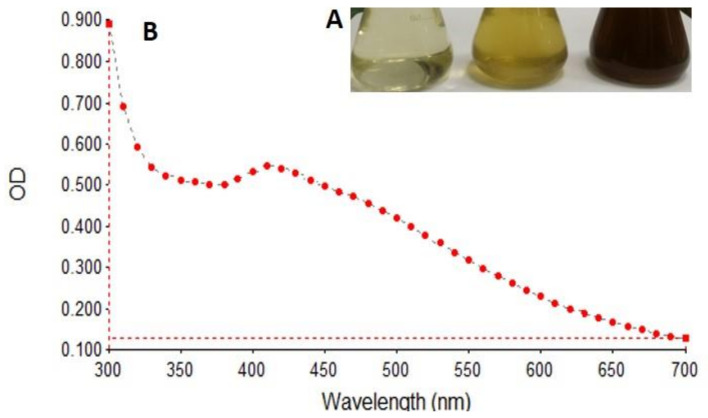
Biosynthesis of silver nanoparticles. Color change reaction in the mixture (*S. molle* extract + silver nitrate) from yellow to brown (**A**). UV-vis spectroscopy scanning at the wavelength of 300–700 nm exhibits intense SPR peak at 412 nm (**B**).

**Figure 2 nanomaterials-12-00161-f002:**
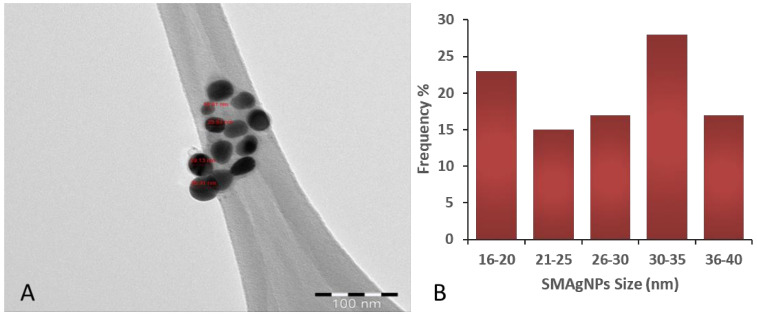
The transmission electron microscopy micrograph of the biosynthesized SMAgNPs (**A**) shows the spherical shape, with size distribution ranging from 16.5–38.5 nm. Quantitative analysis of more than 100 SMAgNPs revealed the frequency at which the NPs were found to be in the above range (**B**).

**Figure 3 nanomaterials-12-00161-f003:**
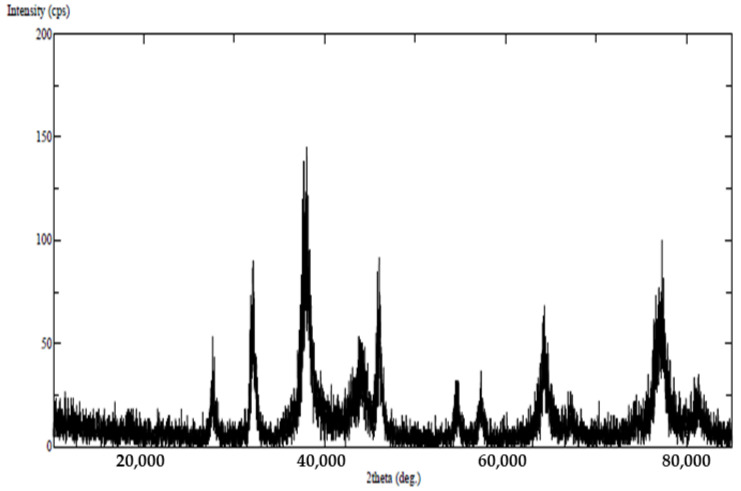
XRD spectrum pattern of biosynthesized AgNPs from the aqueous extract of *S. molle*. The diffraction planes at (111), (200), (220), and (311) reveal the cubic crystalline nature of the SMAgNPs.

**Figure 4 nanomaterials-12-00161-f004:**
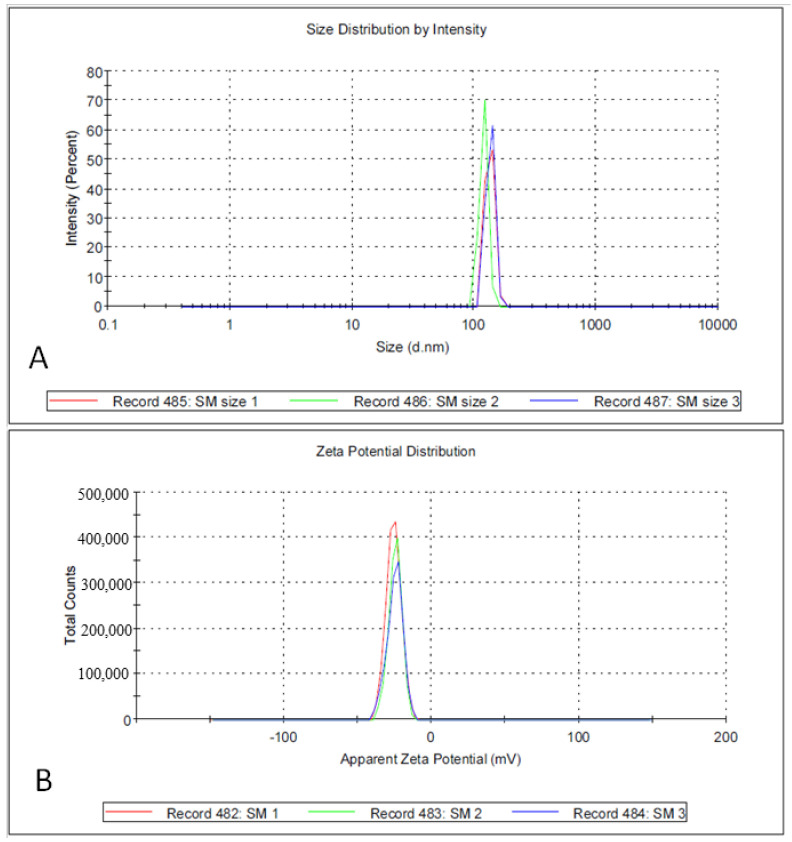
Size distribution analysis of biosynthesized SMAgNPS by dynamic light scattering (DLS) observed as 135.8 nm (**A**). Zeta potential was recorded as −24.9 mV (**B**).

**Figure 5 nanomaterials-12-00161-f005:**
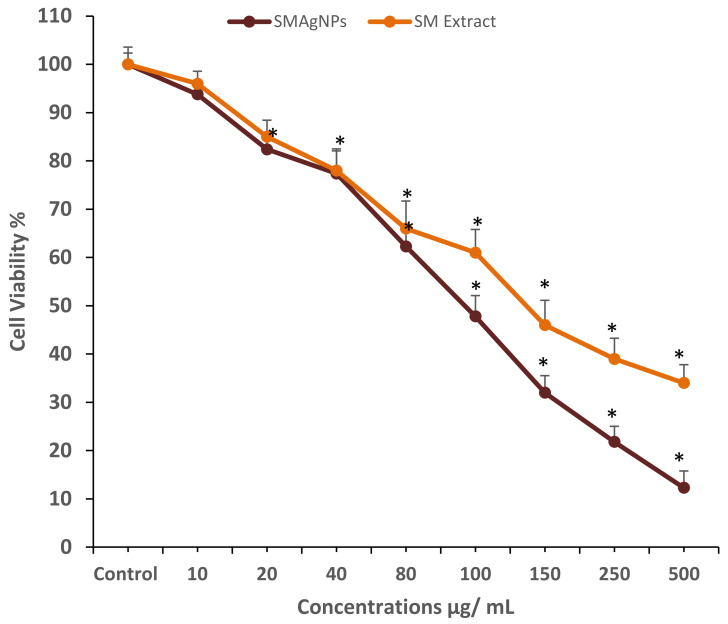
Cytotoxicity of SMAgNPs and *S. molle* extract against HepG2 cells after 24 h of treatment with indicated concentrations evaluated by MTT assay. The IC_50_ values of the SMAgNPs and the *S. molle* extract were estimated to be 92 μg/mL and 135 μg/mL, respectively. Data are represented as mean ± SD from three independent experiments performed in triplicate. (* Significant *p* < 0.05).

**Figure 6 nanomaterials-12-00161-f006:**
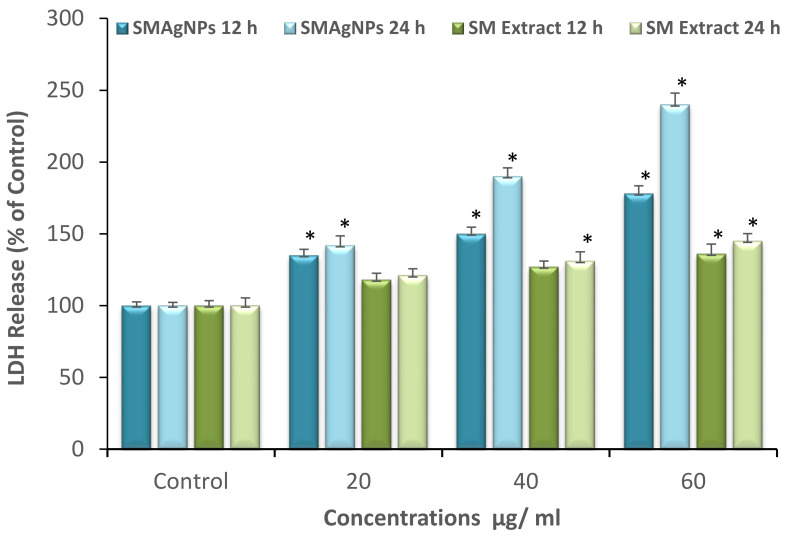
Evaluation of LDH leakage induced in HepG2 cells by different concentrations of SMAgNPs and SM extract in HepG2 cells after 12 h and 24 h of incubation, respectively. Data are represented as mean ± SE from three independent experiments performed in triplicate. (* Significant *p* < 0.05).

**Figure 7 nanomaterials-12-00161-f007:**
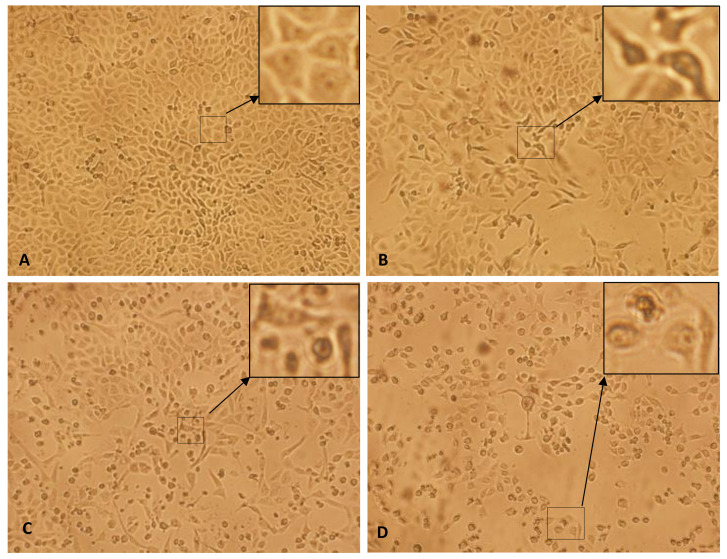
Inverted microscopic images displaying the morphological alterations of HepG2 cells in a time- and dose-dependent manner after exposure to different concentrations of SMAgNPs for 24 h. (**A**) Control cells, (**B**) 20 µg/mL (**C**) 40 µg/mL, (**D**) 60 µg/mL. (Magnification: 100×).

**Figure 8 nanomaterials-12-00161-f008:**
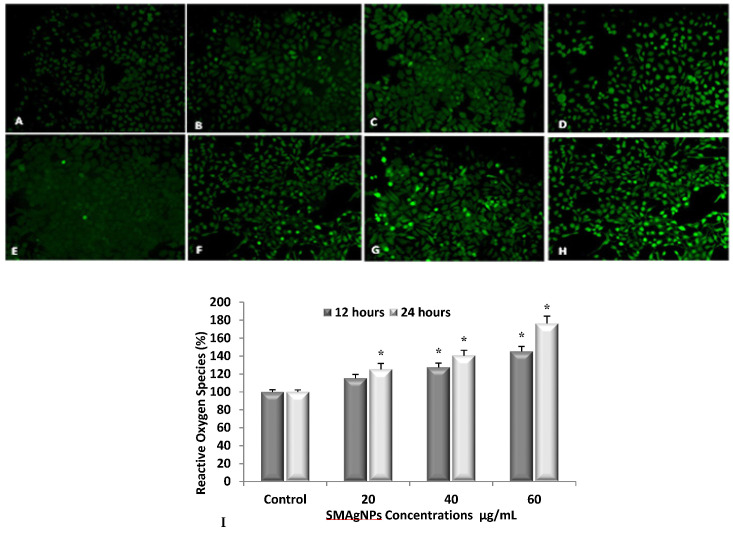
Intracellular ROS generation in HepG2 cells exposed to different concentrations of SMAgNPs for 12 h (**A**–**D**) and 24 h (**E**–**H**) using carboxy-H_2_DCFDA fluorescence marker. Representative images of untreated control cells (**A**,**E**) and cells treated with 20 µg/mL (**B**,**F**), 40 µg/mL (**C**,**G**), and 60 µg/mL (**D**,**H**) of SMAgNPs after 12 and 24 h, respectively (magnification 20×). The quantification of green fluorescence intensity (%) in exposed cells relative to untreated controls (**I**). Data are presented as mean ± SD for three independent experiments performed in triplicate (* significant, *p* < 0.05).

**Figure 9 nanomaterials-12-00161-f009:**
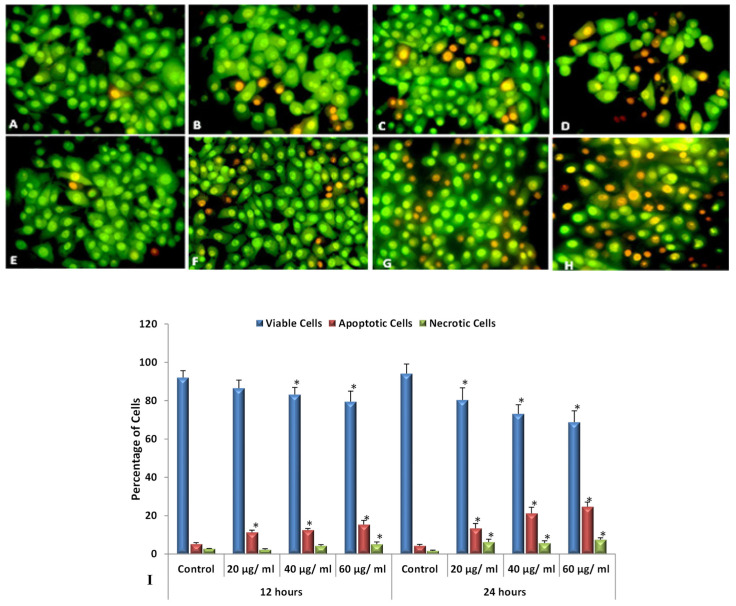
Assessment of apoptotic morphological changes by (acridine orange-ethidium bromide) dual staining method. Representative images of untreated control cells (**A**,**E**) and cells treated with 20 µg/mL (**B**,**F**), 40 µg/mL (**C**,**G**), and 60 µg/mL (**D**,**H**) of SMAgNPs after 12 h (**A**–**D**) and 24 h (**E**–**H**), respectively (magnification 40×). Quantitative evaluation of HepG2 cells revealed apoptotic alteration increased in dose- and time-dependent fashion (**I**). Data are presented as mean ± SD for three independent experiments performed in triplicate (* significant, *p* < 0.05).

**Figure 10 nanomaterials-12-00161-f010:**
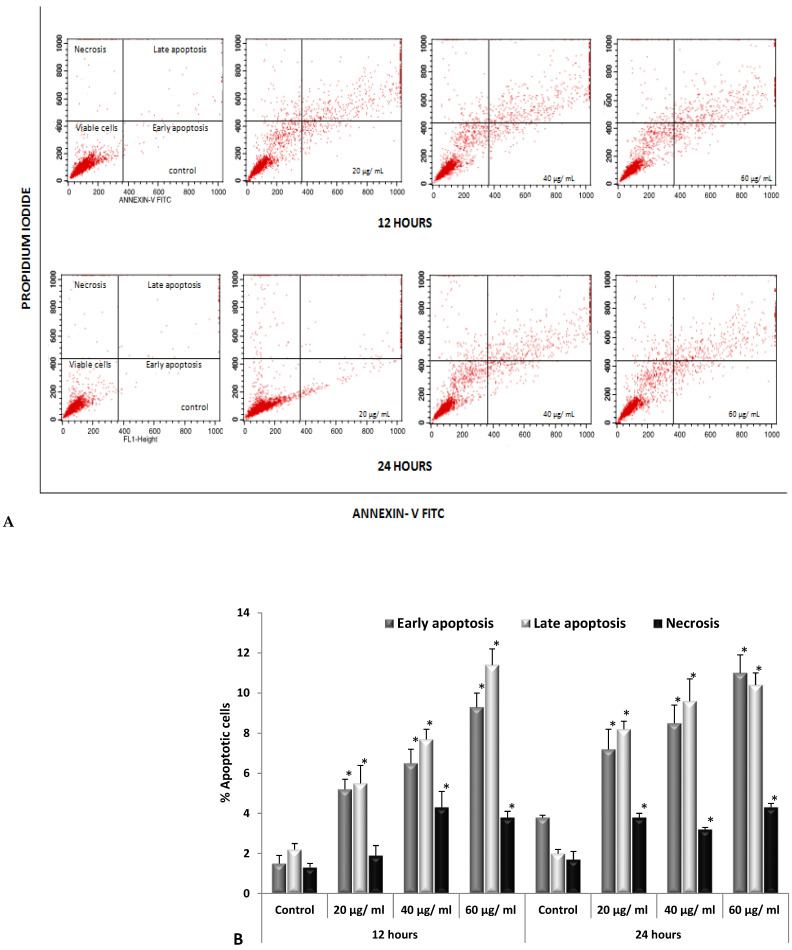
Evaluation of apoptosis using annexin V-FITC/PI assay in HepG2 cells exposed to the different concentrations of SMAgNPs. Dot plots representing percentage of viable cells, early apoptotic cells, late apoptotic cells, and necrotic cells after 12 and 24 h (**A**). Bar diagram showing the percent of apoptotic cells relative to the control (**B**).

**Figure 11 nanomaterials-12-00161-f011:**
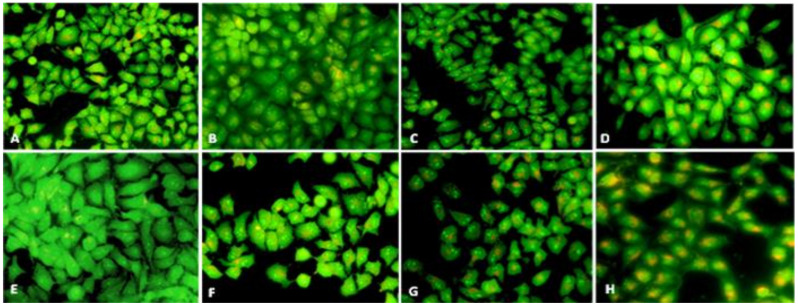
Detection of autophagy triggered by SMAgNPs in HepG2 cells after the treatment using AVOs staining with AO. Representative fluorescent microscopic images of control (**A**,**E**), 20 µg/mL (**B**,**F**), 40 µg/mL (**C**,**G**), and 60 µg/mL (**D**,**H**) after 12 h (**A**–**D**) and 24 h (**E**–**H**), respectively. (Magnification 40×).

**Figure 12 nanomaterials-12-00161-f012:**
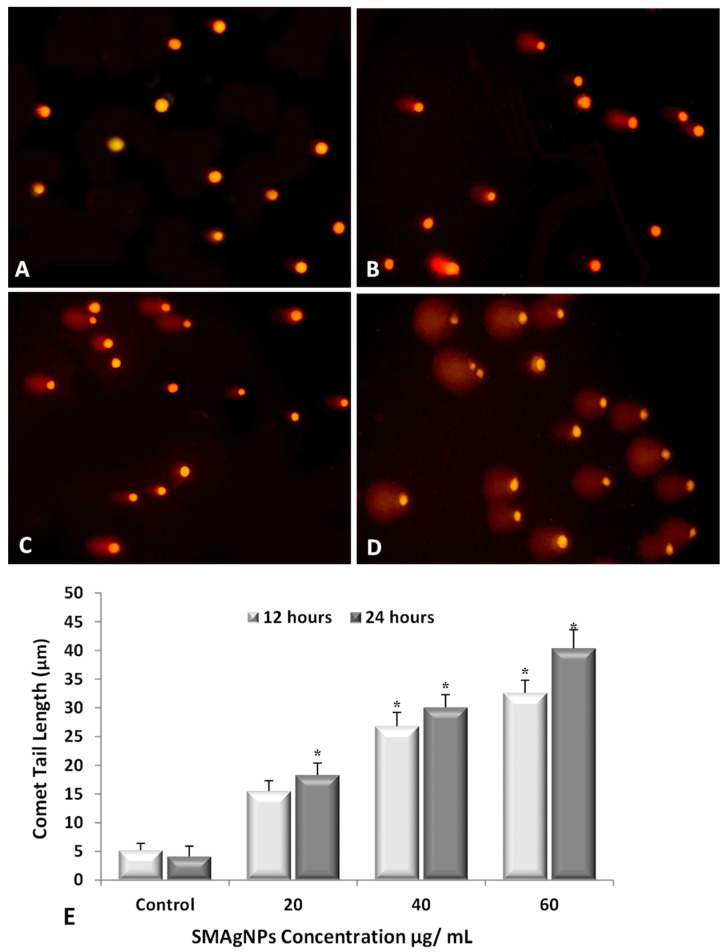
Detection of DNA damage determined by comet assay in HepG2 cells exposed to different concentrations of SMAgNPs for 24 h. The representative images of cells with comet tails, which represent DNA damage (magnification 20×). Control cells (**A**), 20 µg/mL (**B**), 40 µg/mL (**C**), and 60 µg/mL (**D**). The comet tail length (µm) for all treatments is expressed as mean ± SD from three independent experiments (**E**), (* Significant, *p* < 0.05).

## Data Availability

The original contributions presented in the study are included in the article; further inquiries can be directed to the corresponding authors.

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
