# Peer review of "Reactive Oxygen Species-Mediated Cytotoxicity in Liver Carcinoma Cells Induced by Silver Nanoparticles Biosynthesized Using Schinus molle Extract"

_nanomaterials, 2022, doi:10.3390/nano12010161_

Round 1
Reviewer 1 Report
The aim of the present manuscript is very interesting. The authors selected for stabilization of silver nanoparticles S. molle leaf extract because S. molle is known to be used as antitumoral agent. So in combination with silver nanoparticles (which also have antitumoral properties) the combination of two antitumoral agents could give high effectivity and probably synergy. Excellent idea. But for this the authors should use S. molle leaf extract as a second control and to discuss the final results comparing the effectivity of their formulation with effectivity of other silver nanoparticle formulations. Unfortunately, this was not presented in the work.
Major points are:
- Authors claim that “Schinus molle is one of the most important medicinal plant that has been used as antifungal antibacterial, antiviral, topical antiseptic, analgesic, anti-inflammatory antispasmodic, and antioxidant as well as, anti-tumoural agent [13-15].” In this case it was important to study S. molle leaf extract as second control. Probably the cytotoxic effect of the biosynthesized AgNPs using S. molle against the HepG2 liver cancer cells was only due to S. molle leaf extract and not due to silver nanoparticles. Please add data for S. molle leaf extract without silver.
- Lines 176-177. “The process completely finished after 2 h” is a statement without clear evidence. UV–vis spectrum in Fig. 1 shows the sharp increase of the spectrum in the interval < 300 nm, which indicates that the solution still contained silver ions (Ag+), which are characterized by peak at around 200 nm. So, this observed sharp increase of the spectrum in the interval < 300 nm indicates that in final extract solution metallic Ag nanoparticles can coexist with silver cations.
- It is necessary to indicate in Materials and Methods the concentration of metallic silver in silver nanoparticles and the method with which this concentration was measured.
- It is necessary to indicate in Materials and Methods if the concentrations in figures correspond to silver nanoparticles (metallic silver + stabilizer) or to metallic silver.
- The authors should indicate the number of analyzed silver nanoparticles in TEM study in order to assert that their size changes in the interval from 16.5 to 38.5 nm. Figure 2 illustrates the TEM micrograph of only 12 silver nanoparticles. Usually for size distribution the number of analyzed nanoparticles should be ≥ 100.
- Lines 364-365. The phrase “It was also observed that SMAgNPs were well dispersed and stable for long term period.” should be changed.
- What does it mean “well dispersed”? It is better to indicate the average size.
- The manuscript does not provide any evidence of the stability of silver nanoparticles for long period. Please add evidence.
- Lines 395-396. “Notably, AgNPs have been described to contain a ten-times stronger inhibition on HepG2 cells versus normal cells [41].” The authors cannot apply the toxicity values measured for another silver particle formulation for their formulation. The toxicity of different silver particle formulations cannot be generalized. The toxicity for different silver nanoparticle formulations with respect to healthy (normal) cells varies a lot. Every silver nanoparticle formulation is characterized by its own toxicity. Toxicity depends on stabilizing agent, its concentration, etc. For example, 5% hemolysis of human erythrocytes caused by different formulations of silver nanoparticles occurs at silver nanoparticle concentrations differing by more than 40 times. Wherein some “green synthesized” formulations were the most toxic for human erythrocyte. [Roberto Luna-Vázquez-Gómez, et al., Hemolysis of Human Erythrocytes by Argovit™ AgNPs from Healthy and Diabetic Donors: An In Vitro Study, Materials 2021, 14, 2792. https://doi.org/10.3390/ma14112792]
- It is necessary to discuss the comparison of IC50 value obtained by authors (92 μg/ mL) with IC50 obtained for other formulations of silver nanoparticles in the literature.
Minor points:
1.Abstract, line No. 20 and Introduction, line 55: put comma after the word “spectrophotometry“.
- Line 154. Separate “for” and “15” in for15 min
- Homogenize the letter size, font and space between lines for 2. 11. Statistical analysis.
- Fig. 4 and 6. Correct designation of “Standard error”, which actually is written as “+SE”.
- Line 360. Write in Schinus molle by italic.
- Change “anti-tumoural” to “antitumoral”.
Author Response
Answers to the reviewer Comments
Manuscript Title: Reactive Oxygen Species Mediated Cytotoxicity in Liver Carcinoma Cells Induced by Silver Nanoparticles Biosynthesized Using Schinus molle Extract.
Manuscript ID: Nanomaterials-1476177
Reviewer 1
The aim of the present manuscript is very interesting. The authors selected for stabilization of silver nanoparticles S. molle leaf extract because S. molle is known to be used as antitumoral agent. So, in combination with silver nanoparticles (which also have antitumoral properties) the combination of two antitumoral agents could give high effectivity and probably synergy. Excellent idea. But for this the authors should use S. molle leaf extract as a second control and to discuss the final results comparing the effectivity of their formulation with effectivity of other silver nanoparticle formulations. Unfortunately, this was not presented in the work.
Major points are:
- Authors claim that“Schinus molle is one of the most important medicinal plant that has been used as antifungal antibacterial, antiviral, topical antiseptic, analgesic, anti-inflammatory antispasmodic, and antioxidant as well as, anti-tumoural agent [13-15].” In this case it was important to study S. molle leaf extract as second control. Probably the cytotoxic effect of the biosynthesized AgNPs using molle against the HepG2 liver cancer cells was only due to S. molle leaf extract and not due to silver nanoparticles. Please add data for S. molle leaf extract without silver.
Answer: We have performed the cytotoxicity assay (MTT and LDH release) with the S. molle extract (the same extract was used for the synthesis of AgNPs). Please check the revised figure 5 and figure 6. It was observed that the cytotoxicity of SMAgNPs was high as compared to the S. molle extract. From the results, it appears that there were additive effects of phytochemicals present in extract when attached to the AgNPs. (please see the text of the manuscript, Lines: 262-265 and 285-286).
- Lines 176-177. “The process completely finished after 2 h” is a statement without clear evidence. UV–vis spectrum in Fig. 1 shows the sharp increase of the spectrum in the interval < 300 nm, which indicates that the solution still contained silver ions (Ag+), which are characterized by peak at around 200 nm. So, this observed sharp increase of the spectrum in the interval < 300 nm indicates that in final extract solution metallic Ag nanoparticles can coexist with silver cations.
Answer: The above sentence was removed from the text since we have no absolute evidence. This fact could not be ruled out that in final extract solution metallic Ag nanoparticles can coexist with silver cations. It was assumed that after centrifugation at high speed, free phytocomponents and silver ions were removed and only biosynthesized AgNPs were collected.
- It is necessary to indicate in Materials and Methods the concentration of metallic silver in silver nanoparticles and the method with which this concentration was measured.
Answer: The accurate concentration of silver metal in AgNPs could be measured by atomic adsorption spectroscopy. Unfortunately, due to unavailability of this facility, the measurement of the metal concentration in AgNPs was not performed. Nevertheless, we have analyzed particle size and zeta potential through dynamic light scattering technique (Please see Figure 4 in revised manuscript). Moreover, the main objectives of the present study are biosynthesis of AgNPs and its application to the cancer cells after basic characterization.
- It is necessary to indicate in Materials and Methods if the concentrations in figures correspond to silver nanoparticles (metallic silver + stabilizer) or to metallic silver.
Answer: The concentrations were indicated to biosynthesized silver nanaoparticles (SMAgNPs, metallic silver + stabilizer).
- The authors should indicate the number of analyzed silver nanoparticles in TEM study in order to assert that their size changes in the interval from 16.5 to 38.5 nm. Figure 2 illustrates the TEM micrograph of only 12 silver nanoparticles. Usually for size distribution the number of analyzed nanoparticles should be ≥ 100.
Answer: The number of analyzed nanoparticles was more than 100 particles and we have captured several TEM micrographs but a representative one was selected to present in the manuscript that contains the smallest and the largest size particles. Please refer to Figure 2 B where frequency of more than 100 SMAgNPs were presented in bar diagrams.
- Lines 364-365. The phrase “It was also observed that SMAgNPs were well dispersed and stable for long term period.” should be changed.
- What does it mean “well dispersed”? It is better to indicate the average size.
- The manuscript does not provide any evidence of the stability of silver nanoparticles for long period. Please add evidence.
Answer: The above phrase was removed from the text. Well dispersed mean AgNPs are separated in suspension and not aggregated or make cluster. Besides, the biosynthesized AgNPs showed a negligible change in SPR peaks in UV–Vis spectrum after one year of storage at 4 oC (data not shown).
- Lines 395-396. “Notably, AgNPs have been described to contain a ten-times stronger inhibition on HepG2 cells versus normal cells [41].” The authors cannot apply the toxicity values measured for another silver particle formulation for their formulation. The toxicity of different silver particle formulations cannot be generalized. The toxicity for different silver nanoparticle formulations with respect to healthy (normal) cells varies a lot. Every silver nanoparticle formulation is characterized by its own toxicity. Toxicity depends on stabilizing agent, its concentration, etc. For example, 5% hemolysis of human erythrocytes caused by different formulations of silver nanoparticles occurs at silver nanoparticle concentrations differing by more than 40 times. Wherein some “green synthesized” formulations were the most toxic for human erythrocyte. [Roberto Luna-Vázquez-Gómez, et al., Hemolysis of Human Erythrocytes by Argovit™ AgNPs from Healthy and Diabetic Donors: An In Vitro Study, Materials 2021, 14, 2792. https://doi.org/10.3390/ma14112792]
Answer: It is right; the cytotoxic effect of different AgNP formulations on different normal cells or cancer cells varies a lot. In this context, that study was included because it applied the same cancer cell line (HepG2) that were used in the present work. However, we have removed the phrase from the text.
- It is necessary to discuss the comparison of IC50 value obtained by authors (92 μg/ mL) with IC50 obtained for other formulations of silver nanoparticles in the literature.
Answer: In general, the IC50 value of different silver nanoparticles formulations differ widely according to the size of NPs, cell lines type, stabilizing agent present in the extract and its concentrations. A comparative account was included in the results and discussion section (Lines 269 to 273).
Minor points:
1.Abstract, line No. 20 and Introduction, line 55: put comma after the word “spectrophotometry“.
Answer: Corrected
- Line 154. Separate “for” and “15” in for15 min
Answer: Changed
- Homogenize the letter size, font and space between lines for 11. Statistical analysis.
Answer: Formatted section 2.11. Statistical analysis
- 4 and 6. Correct designation of “Standard error”, which actually is written as “+SE”.
Answer: Corrected to ±SE, now changed to ± SD
- Line 360. Write in Schinus molle by italic.
Answer: Done in italics
- Change “anti-tumoural” to “antitumoral”.
Answer: Changed

Reviewer 2 Report
Comments to the Author:
In this manuscript, the author described the synthesis, characteristics, and cytotoxicity evaluation of silver nanoparticles (AgNPs) biosynthesized successfully using leaves extract of S. molle as an active reducing agent. The UV–vis, TEM, and XRD techniques were carried out to characterize the biosynthesized AgNPs. Besides, the cytotoxicity evaluation to HepG2 liver cancer cells, reactive oxygen species generation, apoptosis induction, DNA damage, and autophagy activity were also investigated for the biosynthesized AgNPs. However, there are still many problems in this article, I suggest that the article can be accepted for publication in Nanomaterials after addressing the following comments.
- A series of logical problems was existed in the Introduction. Please carefully review the logical relationship of the manuscript writing and make corrections. For examples, there is a turning point in the last two sentences of the first paragraph. After describing that cancer treatment strategies are used to improve human life, the major problems are introduced immediately. "Besides" should be modified to "However" or similar words.
- The concept of Schinus molle L. should be introduced in the first line of the third paragraph, but the predicate verb is missing, and “Schinus molle L. one of the most important medicinal plant” should be corrected to “Schinus molle L. is one of the most important medicinal plant”.
- The full name should be provided when the abbreviation first appears in the text. For example, “transmission electron microscope” should be added in the front of “TEM”.
- UV-vis and TEM should belong to the same characterization methods and should be separated by conjunctions or commas.
- The concept of IC50 should be described in the manuscript, please add it.
- Serious tense and grammatical problems are present in the manuscript, please review and revise them carefully. For example, the descriptions in methods and results should use the past tense, and the subject and predicate should match each other.
- The figures lack intuitiveness and aesthetics, and further design and beautification should be required. For examples, images with insufficient clarity are layout irregularly.
- It is recommended that the third part (Results) and the fourth part (Discussion) are combined, which can improve the readability and logic of the manuscript.
- There are many problems in grammar, tense, and format. Please carefully check and correct the errors according to the requirements of Nanomaterials.
For example:
(1) The font of "biosynthesis" on line 45 should be changed to the Times New Roman.
(2) The ";" on line 49 should be changed to ",".
(3) The "bio synthesized" on line 71 should be changed to " biosynthesized ".
(4) There should be a space between the number and the unit. The “37℃” should be changed to “37 ℃”.
(5) There are still many similar errors in the manuscript, please correct them.
Author Response
Answers to the reviewer Comments
Manuscript Title: Reactive Oxygen Species Mediated Cytotoxicity in Liver Carcinoma Cells Induced by Silver Nanoparticles Biosynthesized Using Schinus molle Extract.
Manuscript ID: Nanomaterials-1476177
Reviewer 4
Comments to the Author:
In this manuscript, the author described the synthesis, characteristics, and cytotoxicity evaluation of silver nanoparticles (AgNPs) biosynthesized successfully using leaves extract of S. molle as an active reducing agent. The UV–vis, TEM, and XRD techniques were carried out to characterize the biosynthesized AgNPs. Besides, the cytotoxicity evaluation to HepG2 liver cancer cells, reactive oxygen species generation, apoptosis induction, DNA damage, and autophagy activity were also investigated for the biosynthesized AgNPs. However, there are still many problems in this article, I suggest that the article can be accepted for publication in Nanomaterials after addressing the following comments.
- A series of logical problems was existed in the Introduction. Please carefully review the logical relationship of the manuscript writing and make corrections. For examples, there is a turning point in the last two sentences of the first paragraph. After describing that cancer treatment strategies are used to improve human life, the major problems are introduced immediately. "Besides" should be modified to "However" or similar words.
Answer: Corrected the sentences by rephrasing. We have availed language editing services from MDPI. The manuscript is thoroughly checked for any mistakes and error.
- The concept of Schinus molle L. should be introduced in the first line of the third paragraph, but the predicate verb is missing, and “Schinus molle L. one of the most important medicinal plant” should be corrected to “Schinus molle L. is one of the most important medicinal plant”.
Answer: Corrected the sentences by rephrasing.
- The full name should be provided when the abbreviation first appears in the text. For example, “transmission electron microscope” should be added in the front of “TEM”.
Answer: Abbreviations were corrected and incorporated in first appearance.
- UV-vis and TEM should belong to the same characterization methods and should be separated by conjunctions or commas.
Answer: Corrected in line 56 Last paragraph of Introduction
- The concept of IC50 should be described in the manuscript, please add it.
Answer: The concept of IC50 has been incorporated in the text (Lines:109-113 section 2.3)
- Serious tense and grammatical problems are present in the manuscript, please review and revise them carefully. For example, the descriptions in methods and results should use the past tense, and the subject and predicate should match each other.
Answer: The manuscript has been checked thoroughly and grammar, tense and format were corrected by taking help from professional language editors through author services from MDPI.
- The figures lack intuitiveness and aesthetics, and further design and beautification should be required. For examples, images with insufficient clarity are layout irregularly.
Answer: We have tried to improve the clarity of images, although we have presented the best available quality images.
- It is recommended that the third part (Results) and the fourth part (Discussion) are combined, which can improve the readability and logic of the manuscript.
Answer: Results and Discussion section were combined as suggested for more clarity and understanding. Accordingly, references and figures were numbered.
- There are many problems in grammar, tense, and format. Please carefully check and correct the errors according to the requirements of Nanomaterials.
Answer: The manuscript has been checked thoroughly and grammar, tense and format were corrected by taking help from professional language editors through author services from MDPI.
For example:
- The font of "biosynthesis" on line 45 should be changed to the Times New Roman.
Answer: Corrected
- The ";" on line 49 should be changed to ",".
Answer: Changed
- The "bio synthesized" on line 71 should be changed to " biosynthesized ".
Answer: Corrected
- There should be a space between the number and the unit. The “37℃” should be changed to “37 ℃”.
Answer: Corrected at all appearances
- There are still many similar errors in the manuscript, please correct them.
Answer: We have thoroughly checked the whole manuscript for errors and corrected.
Reviewer 3 Report
Comments:
This manuscript “Reactive Oxygen Species Mediated Cytotoxicity in Liver Carcinoma Cells Induced by Silver Nanoparticles Biosynthesized Using Schinus molle Extract” describes anticancer effects of HCG cells in vitro, with the development of SMAgNPs (silver nanoparticles synthesized with Schinus molle extract). Overall, it needs a major revision to be published.
- Please discuss the mechanism how SMAgNPs can generate ROS. Is the mechanism specific to liver cells? If not, how can prevent the non-specific ROS-mediated cytotoxicity to the normal cells when systemically administered to the body? Cytotoxicity test on normal liver cells should be at least required with Figure 4.
- Please analyze mass composition of the SMAgNPs; AgNPs vs. leaf components.
- Please show the time-dependent change of UV-vis spectra in Figure 1.
- In Figure 2, it is unclear whether the SMAgNPs are separated or make clusters. Other images as well as DLS data should be included.
- In Figure 5, enlarged images of cells should be inserted as a inset, in order to see the morphology of the cells in more detail.
- Instead of mean +- standard error of mean, mean +- standard deviation should be employed for in vitro experiments.
- Please check the typo; eg. Figure 6.
- The quantified viable cells in Figure 8 are not correlated with the results in Figure 4.
- Please discuss the reason why autophagy was measured. The description in discussion section is not enough
Author Response
This manuscript “Reactive Oxygen Species Mediated Cytotoxicity in Liver Carcinoma Cells Induced by Silver Nanoparticles Biosynthesized Using Schinus molle Extract” describes anticancer effects of HCG cells in vitro, with the development of SMAgNPs (silver nanoparticles synthesized with Schinus molle extract). Overall, it needs a major revision to be published.
- Please discuss the mechanism how SMAgNPs can generate ROS. Is the mechanism specific to liver cells? If not, how can prevent the non-specific ROS-mediated cytotoxicity to the normal cells when systemically administered to the body? Cytotoxicity test on normal liver cells should be at least required with Figure 4.
Answer: ROS are chemical species that are produced as by-products of cellular oxygen metabolism, which occurs via mitochondrial respiration in eukaryotic cells. Many previous studies have provided strong evidence for a link between AgNP-mediated production of ROS, the subsequent generation of oxidative stress, and cytotoxicity (Asharani et al, 2009; Foldjberg et al, 2009). On the other hand, there is conflicting evidence regarding the dependency of AgNP induced cytotoxicity on ROS (Li et al, 2017). The exact mechanism through which SMAgNPs generate ROS is not clear. Moreover, the mechanism of ROS generation is not specific to the liver cells. However, some of the studies have shown that cancer cells are more sensitive in generating ROS than in normal cells (Bin-Jumah et al, 2020). This is due to the tumor microenvironment which have low pH (acidic) and have high requirement of energy for oxidative metabolism. (Please see the textin the manuscript, Lines: 327-334).
Due to the unavailability of normal liver cell lines, we are unable to perform the cytotoxicity assay.
- Please analyze mass composition of the SMAgNPs; AgNPs vs. leaf components.
Answer: We have followed standard methods to biosynthesize AgNPs and its characterization. Also, our main objective was biological application of the synthesized AgNPs. We have indicated that the concentrations applied in the in vitro experiments for biosynthesized silver nanoparticles was SMAgNPs, (including metallic silver + stabilizer).
Unfortunately, due to some limitations, we were unable to analyze mass composition of the SMAgNPs; AgNPs vs. leaf components.
- Please show the time-dependent change of UV-vis spectra in Figure 1
Answer: During the biosynthesis of AgNPs, UV-vis spectra for time-dependent changes were not observed. The UV-Vis spectra were recorded only at the end of the experiment when the biosynthesis process was completed. The results were presented in figure 1.
- In Figure 2, it is unclear whether the SMAgNPs are separated or make clusters. Other images as well as DLS data should be included.
DLS data and other micrographs of TEM should be added.
Answer: We have captured several TEM micrographs and a representative best one was selected to present in the manuscript that contains the smallest and the largest size particles. Figure 2 B shows frequency of more than 100 SMAgNPs presented in bar diagrams. The analysis of SMAgNPS revealed that they were separated and not made cluster. During the revision, we have analyzed hydrodynamic size, zeta potential and polydispersity index of biosynthesized silver nanoparticles. The data were added in the results section. Pease check revised figure 4 and in the text of the manuscript (lines: 224-228)
- In Figure 5, enlarged images of cells should be inserted as a inset, in order to see the morphology of the cells in more detail.
Answer: as per suggestion, enlarged images of cells were inserted as an inset (please check revised figure 6).
- Instead of mean +- standard error of mean, mean +- standard deviation should be employed for in vitro experiments.
Answer: Standard deviation (mean ± SD) was incorporated in all the results and accordingly diagrams were modified.
- Please check the typo; eg. Figure 6.
Answer: Corrected
- The quantified viable cells in Figure 8 are not correlated with the results in Figure 4.
Answer: Because they were completely different assays and both of them targeted different cellular components and based on different concept. In figure 4 (MTT assay) some early apoptotic cells that still have active oxidoreductase enzymes would calculated as viable cells. MTT assay analyze metabolically active cells while, acridine arrange/ ethidium bromide dual staining analyze apoptotic and necrotic cells. Moreover, the concentrations used in two experiments were different therefore, shown different levels of cell viability.
- Please discuss the reason why autophagy was measured. The description in discussion section is not enough.
Answer: Although, autophagy is linked to pathologic conditions such as cancer, it is being studied as a therapeutic target (Wilde et al., 2018). A number of studies have shown that autophagy is a cellular death mechanism and is a response to various anticancer therapies in many kinds of cancer cells (He and Klionsky, 2009; Kim et al., 2009). There is an increasing body of evidence showing a link between ROS and autophagy, in which ROS generation could trigger autophagy (Scherz-Shouval et al., 2007). (Text incorporated in the manuscript in lines: 395-399).
New references:
- Asharani, P.V.; Low Kah Mun, G.; Hande, M.P.; Valiyaveettil, S. Cytotoxicity and genotoxicity of silver nanoparticles in human cells. ACS Nano 2009, 3, 279–
- Foldbjerg, R.; Olesen, P.; Hougaard, M.; Dang, D.A.; Hoffmann, H.J.; Autrup, H. PVP-coated silver nanoparticles and silver ions induce reactive oxygen species, apoptosis and necrosis in THP-1 monocytes. Lett. 2009, 190, 156–162.
- Li, Y.; Qin, T.; Ingle, T.; Yan, J.; He, W.; Yin, J.J.; Chen, T. Diferential genotoxicity mechanisms of silver nanoparticles and silver ions. Toxicol. 2017, 91, 509–19.
- Bin-Jumah, M.; Monera, A.A.; Albasher, G.; Alarifi, S. Effects of green silver nanoparticles on apoptosis and oxidative stress in normal and cancerous human hepatic cells in vitro. J. Nanomed. 2020,15, 1537.
- Wilde, L.; Tanson, K.; Curry, J.; Martinez-Outschoorn, U. Autophagy in cancer: a complex relationship. J. 2018, 475, 1939-1954.
- Scherz-Shouval, R.; Shvets, E.; Fass, E.; Shorer, H.; Gil, L.; Elazar, Z. Reactive oxygen species are essential for autophagy and specifically regulate the activity of Atg4. EMBO J. 2007, 26, 1749–1760.
Kim, D.K.; Yang, J.S.; Maiti, K.; Hwang, J.I.; Kim, K.; Seen, D.; Ahn, Y.; Lee, C.; Kang, B.C.; Kwon, H.B.; Cheon, J.; Seong, J.Y.A. Gonadotropin-releasing hormone-II antagonist induces autophagy of prostate cancer cells. Cancer Res. 2009, 69, 923–931.
Reviewer 4 Report
The manuscript entitled "Reactive Oxygen Species Mediated Cytotoxicity in Liver Carcinoma Cells Induced by Silver Nanoparticles Biosynthesized Using Schinus molle Extract" deals with anti-cancer activity of biosynthesized silver nanoparticles. Authors used multiple relevant cytotoxicity markers to get more insight into the mechanism of cytotoxicity of biosynthesized silver nanoparticles. The results are clearly presented, and discussion is adequate. Therefore, I suggest only several minor modifications of the manuscript prior the acceptance.
- Have authors analysed hydrodynamic size, zeta potential and polydispersity index of biosynthesized silver nanoparticles?
- There are no scale bars! Authors should add a standard scale bars to all images.
- Results of autophagy should be quantitative or a justification for only visual analysis must be included.
Author Response
Answers to the reviewer Comments
Manuscript Title: Reactive Oxygen Species Mediated Cytotoxicity in Liver Carcinoma Cells Induced by Silver Nanoparticles Biosynthesized Using Schinus molle Extract.
Manuscript ID: Nanomaterials-1476177
Reviewer 3
The manuscript entitled "Reactive Oxygen Species Mediated Cytotoxicity in Liver Carcinoma Cells Induced by Silver Nanoparticles Biosynthesized Using Schinus molle Extract" deals with anti-cancer activity of biosynthesized silver nanoparticles. Authors used multiple relevant cytotoxicity markers to get more insight into the mechanism of cytotoxicity of biosynthesized silver nanoparticles. The results are clearly presented, and discussion is adequate. Therefore, I suggest only several minor modifications of the manuscript prior the acceptance.
- Have authors analysed hydrodynamic size, zeta potential and polydispersity index of biosynthesized silver nanoparticles?
Answer: During the revision, we have analyzed hydrodynamic size, zeta potential and polydispersity index of biosynthesized silver nanoparticles. The data were added in the results section. Pease check revised figure 4 and in the text of the manuscript (lines: 224-228).
- There are no scale bars! Authors should add a standard scalebars to all images.
Answer: There was a problem in software of fluorescence microscope therefore, we were unable to add scale bar in the images. Nevertheless, we have included the magnification at which the images were captured.
- Results of autophagy should be quantitative or a justification for only visual analysis must be included.
Answer: To accurately estimate autophagic, it is essential to determine autophagic flux, which is defined as the amount of autophagic degradation. Monitoring autophagic flux remains complicated even in cultured cells and model organisms, and is currently unfeasible in humans. Microtubule-associated protein light chain 3 (LC3) is most commonly used marker for quantitative estimation of autophagy through western blot detection. which is time-consuming and tedious process. End-point assay results are solely based on quantity of protein.
As a marker of autophagy induction, acidic vesicular organelles (AVOs), which consist predominantly of autophagosomes, and autolysosomes, were analyzed qualitatively by fluorescence microscopy after the cells were stained with acridine orange. Acridine orange is a fluorescent weak base that accumulates in acidic vesicular spaces and fluoresces bright red. The intensity of the red fluorescence is proportional to the degree of AVO formation. However, the qualitative assay is low throughput without automation, but it is less time consuming and could be completed in easy steps and results are simple to interpret. We did not have availability of materials for western blotting therefore, we have performed qualitative assay as an alternative method to detect autophagy.
Round 2
Reviewer 1 Report
I think that in the present form the article can be accepted for publication.
Author Response
No comments from the reviewer
Reviewer 2 Report
Comments to the Author:
In this manuscript, the author described the synthesis, characteristics, and cytotoxicity evaluation of silver nanoparticles (AgNPs) biosynthesized successfully using leaves extract of S. molle as an active reducing agent. The UV–vis, TEM, and XRD techniques were carried out to characterize the biosynthesized AgNPs. Besides, the cytotoxicity evaluation to HepG2 liver cancer cells, reactive oxygen species generation, apoptosis induction, DNA damage, and autophagy activity were also investigated for the biosynthesized AgNPs. However, there are still some problems in this article, I suggest that the article can be accepted for publication in Nanomaterials after addressing the following comments.
- The figures lack intuitiveness and aesthetics, and further design and beautification should be required. For examples, images with insufficient clarity are layout irregularly. The sharpness of images of Figure 8A-H and Figure 9A-H is too low.
- The layout of the Figures in the manuscript needs to be unified, for example Figure 2, Figure 8, Figure 9.
- The format of Reference should be unified. For example, the page number on line 542 is missing.
- There are many problems in tense, and format. Please carefully check and correct the errors according to the requirements of Nanomaterials.
For example:
(1) The "100 SMAgNPs" on line 85 should be added to the unit.
(2) The "BINDER®, Germany" on line 96 should be changed to "BINDER®, Germany".
(3) The "IC50" on line 111 should be changed to " IC50 ".
(4) The font of "S. molle" on line 246 should be changed to Aladdin.
(5) The "Value " on line 266 should be changed to "value".
(6) The "hcompared " on line 407 should be changed to "h compared".
(5) The " 60µg/ mL " on line 411 should be changed to "60 µg/ mL".
Author Response
Answers to the reviewer Comments
Manuscript Title: Reactive Oxygen Species Mediated Cytotoxicity in Liver Carcinoma Cells Induced by Silver Nanoparticles Biosynthesized Using Schinus molle Extract.
Manuscript ID: Nanomaterials-1476177
Comments to the Author:
In this manuscript, the author described the synthesis, characteristics, and cytotoxicity evaluation of silver nanoparticles (AgNPs) biosynthesized successfully using leaves extract of S. molle as an active reducing agent. The UV–vis, TEM, and XRD techniques were carried out to characterize the biosynthesized AgNPs. Besides, the cytotoxicity evaluation to HepG2 liver cancer cells, reactive oxygen species generation, apoptosis induction, DNA damage, and autophagy activity were also investigated for the biosynthesized AgNPs. However, there are still some problems in this article, I suggest that the article can be accepted for publication in Nanomaterials after addressing the following comments.
- The figures lack intuitiveness and aesthetics, and further design and beautification should be required. For examples, images with insufficient clarity are layout irregularly. The sharpness of images of Figure 8A-H and Figure 9A-H is too low.
- Answer: We have presented the best selected images in each category. However, following the instructions to prepare the manuscript in Nanomaterials format, we had to insert the JPEG images into word file. This process resulted in substantial loss of image quality and clarity. We have now prepared the microscopic images in JPEG format with 300 dpi. All these modified images were inserted in the manuscript file and will be uploaded separately with the revised manuscript.
- The layout of the Figures in the manuscript needs to be unified, for example Figure 2, Figure 8, Figure 9.
- Answer: The layout of the figure 8 and figure 9 in the manuscript is modified for better presentation. However, figure 2 have only one image and bar diagram therefore it could not be unified.
- The format of Referenceshould be unified. For example, the page number on line 542 is missing.
- Answer: complete reference included. All the references were checked for complete and correct format.
- There are many problems in tense, and format. Please carefully check and correct the errors according to the requirements of Nanomaterials.
- Answer: The whole manuscript is thoroughly checked for spelling mistake, format and errors were corrected.
For example:
- The "100 SMAgNPs" on line 85 should be added to the unit.
Answer: 100 number of SMAgNPs, added
- The "BINDER®,Germany" on line 96 should be changed to "BINDER®, Germany".
Answer: Changed
- The "IC50" on line 111 should be changed to " IC50".
Answer: IC50 corrected.
(4) The font of "S. molle" on line 246 should be changed to Aladdin.
Answer: Font is corrected and made italics.
(5) The "Value " on line 266 should be changed to "value".
Answer: Corrected.
(6) The "hcompared " on line 407 should be changed to "h compared".
Answer: Corrected.
(5) The " 60µg/ mL " on line 411 should be changed to "60 µg/ mL".
Answer: Corrected.
Reviewer 3 Report
The authors have addressed all my comments with satisfaction, and I hereby recommend acceptance of this paper
Author Response
No comments from the reviewer